# Immune Checkpoint Molecules in Hodgkin Lymphoma and Other Hematological Malignancies

**DOI:** 10.3390/cancers17142292

**Published:** 2025-07-10

**Authors:** Mohamed Nazem Alibrahim, Antonino Carbone, Noor Alsaleh, Annunziata Gloghini

**Affiliations:** 1Department of Internal Medicine, Faculty of Medicine, Zagazig University, Zagazig 44519, Egypt; 20512020101186@medicine.zu.edu.eg; 2Centro di Riferimento Oncologico Aviano, IRCCS, 33081 Aviano, Italy; 3Noor Alsaleh Faculty of Medicine, Cairo University, Giza 12613, Egypt; nour.alsaleh@students.kasralainy.edu.eg; 4Department of Advanced Diagnostics, Fondazione IRCCS Istituto Nazionale dei Tumori, 20133 Milano, Italy; annunziata.gloghini@istitutotumori.mi.it

**Keywords:** immune checkpoint inhibitors, PD-1/PD-L1, CTLA-4, LAG-3, anti-tumor immunity, hematological malignancies, classic Hodgkin lymphoma

## Abstract

Immune checkpoint inhibitors have demonstrated remarkable success in melanoma, non-small-cell lung cancer, and Hodgkin lymphoma; however, their efficacy remains limited in other cancers due to immune resistance mechanisms. Hematological malignancies, such as leukemia, non-Hodgkin lymphomas, and multiple myeloma display unique immune checkpoint dynamics that influence immune checkpoint inhibitor responsiveness. This review explores the mechanisms of immune checkpoint regulation and the clinical applications of immune checkpoint inhibitors in hematological malignancies. It also discusses emerging strategies to overcome immune checkpoint inhibitor resistance, including dual checkpoint blockade and tumor microenvironment modulation.

## 1. Introduction

The immune system plays a crucial role in defending the body against malignancies, in part by regulating T-cell activity through immune checkpoints—key inhibitory pathways that help maintain immune tolerance and prevent autoimmune reactions [1,2,3,4]. However, these regulatory mechanisms are frequently exploited by tumor cells, which upregulate checkpoint ligands to suppress anti-tumor immunity and promote immune evasion [5,6]. Two of the best-known checkpoints are programmed death-1 (PD-1) and cytotoxic T-lymphocyte antigen 4 (CTLA-4), receptors on activated T cells. Binding to their ligands sends “stop” signals that weaken T-cell responses, which creates a tumor-friendly environment [5,7,8]. Additionally, newer checkpoint targets such as lymphocyte activation gene-3 (Lag-3), T-cell immunoglobulin and mucin domain 3 (Tim-3), and T-cell immunoglobulin and ITIM domain (TIGIT) are emerging as clinically relevant therapeutic avenues [4,9]. ICIs, which disrupt these interactions, have shown promise in counteracting tumor immune escape [10].

Classic HL (cHL)—including its nodular sclerosis, mixed cellularity, lymphocyte-rich, and lymphocyte-depleted forms—is one of the blood cancers that is most sensitive to ICIs [11,12,13]. Genetic alterations, particularly the amplification of chromosome 9p24.1, enhance the expression of PD-L1 and PD-L2 on Hodgkin Reed–Sternberg cells, which significantly contributes to immune evasion and sensitivity to PD-1 inhibitors [8]. Figure 1 summarizes the key milestones in and the clinical development of immunotherapy.

In contrast to cHL, NHL has shown variable and generally less robust responses to ICIs compared to cHL, likely due to differences in tumor-infiltrating immune cell composition and PD-L1 expression [13]. However, encouraging outcomes have been observed in certain NHL entities such as primary mediastinal B-cell lymphoma (PMBCL), which shares molecular characteristics with cHL [14].

The therapeutic potential of ICIs extends to AML, especially when they are combined with hypomethylating agents (HMAs) like azacitidine, which offers new avenues for managing relapsed or refractory cases [4,15]. Conversely, early optimism for the use of checkpoint inhibitors in MM has diminished after disappointing outcomes in large-scale clinical trials. The lack of efficacy of ICIs in MM may be partly due to genetic polymorphisms, particularly within the PD-1 and CTLA-4 genes. Specific polymorphisms, like the CTLA4 rs231775 AA/AG genotype, have been linked to poorer outcomes, which highlights the need to identify subgroups of MM patients who could potentially benefit from ICI therapies [13,16,17].

Such variability in therapeutic responses across different hematologic malignancies underscores the need for deeper insights into immune regulation, TME dynamics, and reliable predictive biomarkers. This review discusses the physiological functions of immune checkpoints, evaluates their clinical significance, and addresses the unique challenges and unresolved therapeutic questions in various hematological cancers.

## 2. Mechanisms of Immune Checkpoint Regulation

Figure 2 visually represents the immune checkpoint pathways.

### 2.1. PD-1/PD-L1 Axis

The PD-1/PD-L1 axis is essential for maintaining immune tolerance by regulating T-cell activity in both normal and disease states. It also plays a key role in tumor immune evasion. Under normal physiological conditions, PD-1 (CD279) and PD-L1 (CD274 or B7-H1) act as critical immune regulators. When PD-1 binds to PD-L1, the interaction suppresses T-cell activation, helping prevent the immune system from attacking the body’s own tissues and thus maintaining immune homeostasis. For a T cell to become fully activated, two signals are required. First, antigen-presenting cells (APCs) detect and process pathogens like bacteria, then display fragments of these invaders on their surface via major histocompatibility complex (MHC) molecules. T cells recognize these fragments through their T-cell receptors (TCRs) [18,19]. However, to fully activate T cells, co-stimulatory molecules such as CD80 and CD86 on APCs must bind to the CD28 receptors on T cells. This interaction enables T cells to effectively target and destroy infected cells. Conversely, inhibitory molecules like CTLA-4 and PD-1/PD-L1 limit excessive T-cell activity under normal conditions, protecting the body against inflammation and autoimmunity. Cancer cells, however, can exploit these inhibitory mechanisms to evade immune surveillance and destruction [20,21]. The mechanisms controlling PD-1/PD-L1 expression remain incompletely understood. It is hypothesized that cellular signals and inflammatory factors within the tumor microenvironment (TME) may play a key regulatory role [22].

PD-1 is highly expressed in activated T and B cells, NK cells, myeloid cells, monocytes, neutrophils, dendritic cells, and tumor-infiltrating lymphocytes. Its expression is dynamically influenced by antigenic stimulation, inflammatory cues, and the cellular context within the TME [10,22,23,24].

During initial immune responses, antigen recognition by TCR triggers reversible demethylation at PD-1 regulatory regions (CR-B and CR-C), which enhances its transcription [25]. This activates PD-1 transcription through transcription factors such as NF-κB [26], NFATc1 [27], and STAT3/4 [28]. During acute infections, once the antigens have been cleared, the TCR signaling decreases, enhancer regions undergo remethylation, and the expression of PD-1 is subsequently downregulated [29,30].

In contrast, persistent antigen exposure, such as in chronic infections or cancer, leads to continuous TCR signaling, sustained demethylation, and upregulated PD-1 transcription, which contributes to T-cell dysfunction and immune exhaustion [31,32]. This immune suppression facilitates tumor immune escape. Beyond TCR signaling, other regulatory pathways contribute to PD-1 expression. FoxO1 binds to CR-C to upregulate PD-1 and inhibit PI3K/AKT/mTOR signaling, creating a feedback loop that sustains immune inhibition [32]. Type I interferons (e.g., IFN-α) induce PD-1 via the JAK/STAT pathway and the ISGF3 complex [33]. Additionally, TGF-β1 increases the PD-1 and CTLA-4 levels via the calcium-regulated phosphatase nuclear factor of activated T cells 1 (CaN/NFATc1) pathway, aiding tumor immune evasion [34].

Growing evidence shows that tumors promote T-cell exhaustion via PD-1 signaling. Inhibiting this pathway can partially restore T-cell function and reinvigorate immune responses, and thereby improves the efficacy of cancer immunotherapy [35,36,37].

PD-L1, a 33-kDa glycoprotein of the B7 family, features Ig and IgC extracellular domains [38]. PD-L1 is generally expressed by immune cells such as macrophages, activated T and B cells, dendritic cells, and certain epithelial cells, especially during inflammation [39], whereas PD-L2 serves as an inhibitory ligand that modulates PD-L1 function. PD-L1 also interacts with CD80 on activated T cells [40,41]. Tumor cells adaptively express PD-L1, correlating with immune-rich microenvironments characterized by CD8+ T cells, Th1 cytokines, and interferon signatures [42,43].

The expression of PD-1 and PD-L1 varies significantly across different hematological malignancies. For instance, PD-L2 shows high expression in mantle cell lymphoma [44]. Meanwhile, PD-1 is prominently expressed on T-cells in angioimmunoblastic lymphoma, and the associated follicular dendritic cells often express PD-L1 [45]. PD-1 expression is also documented in T-cells surrounding tumor cells in nodular lymphocyte-predominant HL and on CD4+ T-cells in HTLV-1-associated adult T-cell leukemia and lymphoma [46,47]. In cHL, elevated PD-L1 expression on tumor cells and increased PD-1 expression on infiltrating T cells are associated with impaired immune responses. Blocking PD-1 signaling decreases SHP-2 phosphorylation, restores IFN-γ production, and reverses T-cell exhaustion. This underscores the therapeutic promise of targeting the PD-1 pathway in cHL [48].

Furthermore, in AML, the co-expression of PD-1 and Tim-3 on CD8+ T cells correlates with disease progression [49]. PD-L1 on cancer cells has also been demonstrated to increase following antigen-specific T-cell apoptosis in vitro [5]. In chronic myeloid leukemia, PD-1/PD-L1 is highly expressed by immunosuppressive myeloid-derived suppressor cells (MDSCs), which aids in immune evasion, and PD-L1 blockade increases IL-2 release but does not restore full T-cell proliferation [50].

### 2.2. CTLA-4 Pathway

Cytotoxic T-lymphocyte antigen 4 (CTLA-4) (CD152) is a pivotal immune checkpoint receptor that functions as a negative regulator of T-cell activation through multiple tightly controlled mechanisms. Biologically, CTLA-4 competes with the costimulatory receptor CD28 for the ligands CD80 and CD86, binding them with significantly higher affinity and avidity, thereby attenuating CD28-driven T-cell activation [51,52,53,54]. This interaction occurs at the immunological synapse, where CTLA-4 limits the access of CD28 to its ligands and actively removes CD80/CD86 from the surface of APCs via a process termed transendocytosis, a dominant, cell-extrinsic mechanism that is primarily mediated by regulatory T cells (Tregs) [55,56,57]. CTLA-4 is constitutively expressed at high levels on Tregs and is also induced on the surface of activated conventional T lymphocytes, where it contributes to immune suppression and tolerance [58].

The suppressive effect of CTLA-4 is further enhanced by its highly dynamic intracellular trafficking, which is controlled by proteins such as AP-2, LRBA, and the TRIM/LAX/Rab8 complex, which modulate its recycling and degradation, ultimately regulating its surface expression and immune inhibitory capacity [57,59,60,61,62]. Functionally, CTLA-4 inhibits T-cell responses by recruiting phosphatases to its intracellular domain, reducing IL-2 production, inducing cell cycle arrest, and suppressing T-cell proliferation and differentiation [63].

In hematologic malignancies, dysregulation of this checkpoint has pathological implications. Recurrent activating mutations in CD28 and CD28-CTLA-4 fusion proteins have been identified in T-cell leukemias and lymphomas, and these increase the ligands’ affinity and amplifying costimulatory signals that promote tumor immune escape [64,65].

Therapeutically, anti-CTLA-4 monoclonal antibodies, such as ipilimumab and tremelimumab, enhance anti-tumor immunity by blocking inhibitory CTLA-4 signals, and thereby restore T-cell effector function and increase CD28-mediated costimulation [66,67,68,69,70]. However, this comes at the cost of immune-related adverse events (irAEs), which affects multiple organ systems and reflects the physiological role of CTLA-4 in restraining autoreactive T cells [71,72]. Notably, CTLA-4 blockade expands the diversity of the peripheral TCR repertoire, allowing for polyclonal T-cell expansion and activation, which is associated with both a therapeutic benefit and irAE development [73,74,75]. This therapeutic effect stems from the relief of CTLA-4-mediated inhibition, which allows for the reactivation of T-cell proliferation and promotes their differentiation into cytotoxic T lymphocytes (CTLs), enhancing anti-tumor immunity [76,77]

Additionally, genetic deficiencies in CTLA-4 or its trafficking regulator LRBA result in profound immune dysregulation, autoimmunity, and increased susceptibility to malignancy, which further underscores the checkpoint’s essential regulatory role [78,79,80]. These findings highlight CTLA-4 as a central molecular switch in immune homeostasis and a potent target for immunotherapy, particularly in hematologic contexts where immune evasion plays a key role in disease progression.

### 2.3. LAG-3

Lymphocyte activation gene-3 (LAG-3), also known as CD223 FDC, is an inhibitory transmembrane protein that is expressed on a wide range of immune cells, including effector cells and Tregs, NK cells, activated B cells, and plasmacytoid dendritic cells [81]. LAG-3 expression on activated T cells can be induced by interleukin-2 (IL-2) and IL-12 [9]. Structurally, LAG-3 comprises eight exons, encoding a protein composed of extracellular, transmembrane, and intracellular domains that shares significant homology with CD4 [82]. Its primary ligand is MHC-II, which binds to the D1 domain of LAG-3 with a high affinity that surpasses that of CD4 [83]. This interaction has been shown to suppress the proliferation, activation, cytokine secretion, and cytotoxic functions of both CD4+ and CD8+ T lymphocytes, although the precise contribution of MHC-II to LAG-3’s immunosuppressive activity is still under investigation [84,85].

Among the emerging LAG-3-targeting therapeutics, relatlimab, a first-in-class human IgG4 monoclonal antibody, has demonstrated the ability to restore T-cell function through LAG-3 inhibition [86]. Notably, LAG-3 is frequently co-expressed with PD-1 on CD4+ and CD8+ tumor-infiltrating T cells, and their combined blockade has shown enhanced immunotherapeutic efficacy [87,88].

Recent studies have identified additional LAG-3 ligands, including galectin-3 (Gal-3), fibrinogen-like protein 1 (FGL1), and liver sinusoidal endothelial cell lectin (LSECtin), expanding our understanding of its regulatory potential. Gal-3, expressed in various cancers and activated T cells, binds to LAG-3 to suppress CD8+ T-cell cytotoxicity [84,85,89]. Elevated levels of FGL-1 in cancer cells have been linked to unfavorable clinical outcomes [81,90].

Despite these advances, the full mechanistic scope of LAG-3’s function remains incompletely understood. Deeper insights into its molecular pathways are essential not only to refine existing therapies but also to identify predictive biomarkers and select patients who are most likely to benefit from LAG-3-targeted treatments. Unlocking the complexities of LAG-3 signaling may pave the way for more precise and effective strategies in hematologic cancer immunotherapy.

### 2.4. Other Checkpoints

Additional immune checkpoints include T-cell immunoglobulin and mucin domain 3 (TIM-3). TIM-3 is an inhibitory immune checkpoint receptor that is commonly co-expressed with PD-1 on tumor-infiltrating T cells, as well as on various innate immune cells such as monocytes/macrophages, NK cells, and dendritic cells. Its known ligands include galectin-9 (GAL-9) and carcinoembryonic antigen-related cell adhesion molecule 1 (CEACAM1), both of which trigger downstream inhibitory signaling pathways that lead to T-cell exhaustion or apoptosis [91,92]. Additional ligands include high-mobility group protein B1 (HMGB1) and phosphatidylserine (PtdSer). The interaction between TIM-3 and its ligands can lead to different biological outcomes depending on the cellular context [93]. Due to its frequent co-expression with PD-1, TIM-3 has been implicated as a potential mechanism of resistance to PD-1-targeted therapies and is currently under investigation as a therapeutic target across multiple malignancies, including hematologic cancers [94,95]. Sabatolimab was the first anti-TIM-3 monoclonal antibody to be developed, targeting the Gal-9 and PtdSer binding sites, and was initially developed for use in solid tumors. Currently, over 33 anti-TIM-3 antibodies are under clinical investigation as potential cancer immunotherapies [96].

TIGIT (T-cell immunoglobulin and ITIM domain), also known as Vstm3, VSIG9, or WUCAM, is a co-inhibitory transmembrane receptor that is primarily expressed on CD4+, CD8+, Tregs, and NK cells. It interacts with several ligands, including CD155, CD112, CD113, Fab2, and Nectin-4, which are found on both APCs and tumor cells, with CD155 being its highest-affinity binding partner [97,98,99]. Additionally, DNAM-1 (CD226) is an activating receptor on cytotoxic lymphocytes that promotes tumor-cell recognition and elimination by binding to ligands CD155 and CD112. However, this immune response is counteracted by the inhibitory receptors TIGIT and CD96, which also bind CD155 and suppress DNAM-1-mediated activity [100]. CD112 can similarly interact with CD112R, a co-inhibitory receptor on T cells, further interfering with the function of DNAM-1 [101]. In hematologic malignancies, TIGIT has emerged as a key immune checkpoint. Dysfunctional T-cell subsets marked by low DNAM-1 expression and the co-expression of PD-1 and TIGIT are associated with poor outcomes in AML [102]. In MM, TIGIT is the most upregulated checkpoint compared to PD-1, CTLA-4, LAG-3, and TIM-3, and TIGIT+ T cells are also dysfunctional [103].

While the precise mechanisms are still being investigated, TIGIT is known to suppress T-cell activity by downregulating TCR signaling, outcompeting the activating receptor CD226 for ligand binding, and enhancing regulatory T-cell-mediated immunosuppression. Due to its role in immune evasion, TIGIT has emerged as a promising therapeutic target in hematologic malignancies [104]. More than 45 TIGIT inhibitors have been developed, with most of them being used in the treatment of solid tumors and leukemia. Among them, IBI939 and tiragolumab are notable anti-TIGIT monoclonal antibodies that block the interaction between TIGIT and its primary ligand, CD155 [105,106].

### 2.5. Immune-Related Adverse Events Associated with Immune Checkpoint Inhibitors

Chronic irAEs from ICIs are increasingly recognized as a significant clinical challenge, persisting for months or even years beyond therapy discontinuation [107]. Unlike acute irAEs, which often resolve with corticosteroids, chronic irAEs can lead to long-term dysfunction in multiple organ systems. Endocrine irAEs are the most frequent and persistent, with 83.1% of endocrine toxicities becoming chronic, notably hypothyroidism (85.7%), hypophysitis (100%), and adrenal insufficiency (80%) [108]. Rheumatologic toxicities are the next most common, with 46.3% becoming chronic, particularly inflammatory arthritis and xerostomia [109]. Chronic gastrointestinal irAEs, including colitis and hepatitis, occur in 16.5% of GI cases, but generally have a higher rate of resolution [108]. Neurologic events, such as neuropathies and myasthenia gravis, become chronic in 75% of cases, often with significant morbidity [108,109]. Pulmonary irAEs like pneumonitis persist in 50% of affected patients, sometimes with lasting imaging abnormalities or fibrosis [108,110]. Chronic irAEs may arise from irreversible tissue damage or ongoing low-grade inflammation, and their development is not clearly linked to the class or dose of the ICIs that are used. Management often includes long-term hormone replacement (for endocrine irAEs), immunosuppressants, and multidisciplinary care. Despite being mostly low-grade (96.4% grade 1–2), their long-term impact underscores the need for better preventive strategies and biomarkers [109].

## 3. Clinical Applications of Immune Checkpoint Inhibitors

### 3.1. Hodgkin Lymphoma

Classic HL is characterized by the presence of Hodgkin Reed–Sternberg (HRS) cells surrounded by a rich inflammatory infiltrate, which paradoxically fails to mount an effective anti-tumor immune response. A growing body of evidence suggests that this immune evasion is mediated, in part, by dysregulation of the PD-L1 and PD-L2 signaling axis. Genetic alterations such as the amplification of chromosome 9p24.1, which harbors the genes encoding PD-L1 and PD-L2, have been associated with impaired immune surveillance and poorer clinical outcomes in CHL patients [111,112].

These findings provided the rationale for targeting the PD-1/PD-L1 pathway in relapsed/refractory cHL, leading to the accelerated FDA approval of the PD-1 inhibitors nivolumab [8] and pembrolizumab [113] in 2016 and 2017, respectively. Long-term follow-up data from the KEYNOTE-087 and CheckMate-205 trials have demonstrated durable responses, with objective response rates of over 70% and encouraging overall survival outcomes despite modest progression-free survival (PFS) metrics [8,113,114,115]. These results prompted further investigation into PD-1 blockade in cHL, culminating in the phase 3 KEYNOTE-204 trial, which showed superior PFS with pembrolizumab (13.2 months) compared to brentuximab vedotin (8.3 months) in patients who were ineligible for autologous stem-cell transplant [116].

Due to the efficacy of PD-1 inhibitors in relapsed cHL, (Table 1) several studies were initiated that combined nivolumab or pembrolizumab with other agents as second-line therapy, aiming to enhance patients’ CR rates before consolidative autologous stem-cell transplant (ASCT). This is because achieving CR before ASCT is strongly linked to better outcomes [117].

While combined chemotherapy and radiotherapy effectively treat early-stage cHL, the outcomes remain suboptimal for 10–15% of patients with high-risk features (e.g., B symptoms, bulky disease). Omitting radiotherapy has been consistently linked to worse PFS. In advanced-stage cHL, frontline regimens like ABVD and BEACOPP cure most patients, but 20–30% still experience relapse, and BEACOPP is associated with notable short- and long-term toxicities.

However, the strong efficacy of PD-1 inhibitors, both alone and with chemotherapy, in R/R cHL has driven interest in evaluating their use as part of initial treatment in newly diagnosed patients.

The NIVAHL study [123] evaluated two strategies that combined nivolumab with AVD in patients with early-stage cHL. Among 109 patients, both the concomitant and sequential arms showed high response rates. Interim objective responses were seen in 100% of the patients who received N-AVD and 96% who received nivolumab monotherapy, with CR rates of 87% and 51%, respectively. After completing the therapy, CR was achieved in 90% of the patients in the concomitant arm and 94% in the sequential arm. With a median follow-up of 13 months, the 12-month PFS was 100% and 98%, respectively.

The phase 2 KEYNOTE-C11 [124] trial evaluated a novel sequential approach in previously untreated patients with early unfavorable or advanced-stage cHL, combining pembrolizumab induction, AVD chemotherapy, and pembrolizumab consolidation without radiotherapy. Among 146 enrolled patients, the PET3-negativity rate, the primary endpoint, was 70% in the overall population and 78% in the actuarial cohort, with an ORR of 88% being obtained at this stage. Despite modest PET2-negativity rates (29–31%), the responses deepened with subsequent AVD. The regimen demonstrated manageable toxicity, with grade ≥ 3 adverse events being reported in 16% of patients during pembrolizumab treatment, 69% during AVD treatment, and 59% during escalated BEACOPP treatment (used for PET3-positive patients). Immune-related events occurred in 25%, most commonly thyroid dysfunction. These findings support the safety and efficacy of the chemotherapy-sparing, response-adapted strategy and provide a rationale for the further investigation of PD-1 blockade in the frontline setting.

The randomized phase 3 S1826 [125] trial compared nivolumab plus AVD (N-AVD) to brentuximab vedotin plus AVD (BV-AVD) in 976 patients with newly diagnosed advanced-stage cHL. With a median follow-up of 12.1 months, the PFS was significantly improved in the N-AVD arm (1-year PFS: 94% vs. 86%). Further, < 1% of patients in either arm received radiotherapy. The frequency of grade ≥ 3 hematologic toxicity was higher with N-AVD (48.4% vs. 30.5%), particularly neutropenia, although the rates of febrile neutropenia and non-hematologic adverse events were similar. Peripheral neuropathy was more common with BV-AVD, while thyroid dysfunction was more frequent with N-AVD. These results demonstrate that N-AVD is a more effective frontline regimen than BV-AVD in advanced-stage cHL, with an acceptable safety profile and minimal reliance on radiotherapy.

In a recent phase II trial [126], tislelizumab, an engineered anti-PD-1 antibody designed to minimize Fcγ receptor binding, demonstrated a high overall ORR (87.1%) and CR rate (67.1%) in 70 patients with R/R cHL who had failed or were ineligible for ASCT. At a median follow-up of 33.8 months, the median PFS reached 31.5 months and the 3-year OS rate was 84.8%, indicating a durable clinical benefit. The treatment was well tolerated, with a manageable safety profile. However, the study was limited by its single-arm design, small sample size, and lack of a comparator group, which restricts its generalizability and precludes direct efficacy comparisons.

As PD-1 inhibitors are increasingly used in the frontline treatment of cHL, relapse management strategies must evolve accordingly. Salvage regimens that incorporate checkpoint inhibitors remain untested in patients who have been previously exposed to nivolumab or pembrolizumab, although retreatment may benefit those who were not initially refractory. Identifying relapsed patients who can be cured without ASCT remains a key objective, but the alternative treatment must be tailored to prior therapies and the response duration. Notably, patients with primary refractory disease or early relapse after undergoing PD-1-based frontline therapy will likely still require ASCT for durable remission. Future strategies should aim to better integrate PD-1 inhibitors across treatment lines, which could potentially reduce reliance on chemotherapy, radiation, and transplant.

### 3.2. Non-Hodgkin Lymphoma

NHL, predominantly that of B-cell origin, is often characterized by an immunosuppressive tumor microenvironment driven by intratumoral CD4+CD25+ Tregs that inhibit effector T cell infiltration [127]. These Tregs express high levels of immune checkpoints like CTLA-4, which has prompted investigation into CTLA-4 blockade in clinical trials. In a phase 1 study of ipilimumab, an anti–CTLA-4 monoclonal antibody, in R/R B-cell NHL, unfortunately, only 2 of 18 patients (11%) showed clinical responses [128]. Combining ipilimumab with rituximab resulted in a modest overall response rate (ORR) of 24% and a median PFS of 2.6 months [129]. Furthermore, dual CTLA-4/PD-1 blockade with ipilimumab and nivolumab achieved an ORR of 19%, with a median PFS of only 1 month [130] (Table 2).

**Table 2 cancers-17-02292-t002:** Immune checkpoint inhibitors in non-Hodgkin lymphoma.

Regimen	Population	Checkpoint Target	PFS	CR	Side Effects	Reference
Nivolumab 3 mg/kg every 2 weeks	Patients with R/R DLBCL ineligible for or who failed ASCT	PD-1	Median PFS: 1.9 months (ASCT failed), 1.4 months (ASCT ineligible)	3% (ASCT-failed cohort only)	Grade 3–4 AEs in 24% of patients: neutropenia (4%), thrombocytopenia (3%), increased lipase (3%)	[131]
Nivolumab 3 mg/kg every 2 weeks	Patients with R/R FL (≥2 prior lines, including CD20 antibody and alkylating agent)	PD-1	Median PFS: 2.2 months (95% CI: 1.9–3.6)	1%	TRAEs in 54%, with Grade 3–4 TRAEs in 15%. Most common: fatigue (13%), diarrhea (11%), nausea (10%). Grade 3–4 AEs: neoplasm progression (5%), neutropenia (5%), abdominal pain (4%), anemia (4%). Serious immune-related AEs: pneumonitis, rash, colitis, and toxic epidermal necrolysis (each ~1%). 3 treatment-related deaths occurred	[132]
Durvalumab + R-CHOP (Arm A) Durvalumab + R2-CHOP (Arm B)	Previously untreated, high/high-intermediate risk DLBCL (IPI ≥ 3/NCCN-IPI ≥ 4)	PD-L1	68% (Arm A) and 67% (Arm B) progression-free at 12 months	54% (Arm A); 67% (Arm B)	Fatigue (61%), neutropenia (52%), neuropathy (50%), nausea (46%), diarrhea (28%). Grade 3–4 AEs in 84% (A) and 100% (B). No treatment-related deaths.	[133]
Ipilimumab (3 mg/kg loading, then 1–3 mg/kg monthly × 3–4 months)	R/R B-cell lymphoma (DLBCL and FL)	CTLA-4	-	1 CR (DLBCL, >31 months); 1 PR (FL, 19 months)	Generally well tolerated	[128]
Rituximab + Ipilimumab	R/R CD20+ B-cell NHL (including FL)	CTLA-4	Median PFS: 2.6 months overall; 5.6 months in FL subset	Not specified (ORR 58% in FL)	Manageable toxicity. No dose-limiting toxicities. Common AEs not detailed	[129]
Nivolumab + Ipilimumab	R/R B-NHL (DLBCL, FL)	PD-1 + CTLA-4	Median PFS: 1–2 months	6% (FL: 0%; DLBCL: 9%)	TRAEs occurred in up to 79% of NHL patients, with grade 3–4 events in 15–29%. Common AEs included skin toxicity, fatigue, diarrhea, fever, and infusion reactions. Serious AEs (~3–5%) included pneumonitis, neutropenia, tumor flare, and autoimmune complications. Discontinuation occurred in 8% (nivo/ipi); none with nivo/liri. No treatment-related deaths.	[130]
Nivolumab + Lirilumab	R/R B-NHL (DLBCL, FL)	PD-1 + KIR	Median PFS: 1–2 months	3% (FL: 17%; DLBCL: 0%)		
Nivolumab + Ipilimumab/Lirilumab	R/R T-NHL	PD-1 + CTLA-4 or KIR	Median PFS: ~6 months (liri); ~1–2 mo (ipi)	0% (both combinations)	tolerability acceptable.	
Nivolumab + Ipilimumab/Lirilumab	R/R MM	PD-1 + CTLA-4 or KIR	Median PFS: ~1 month	0%	No clinical activity in MM; no unexpected toxicity.	
Pembrolizumab 200 mg every 3 weeks	R/R PMBCL, post ≥ 2 lines of therapy, including patients ineligible for ASCT	PD-1	Median PFS: 4.3 months; 4-year PFS rate: 33.0%	20.8%	56.6% had treatment-related AEs; most common: neutropenia (18.9%), asthenia (9.4%), hypothyroidism (7.5%), fatigue (5.7%). Grade 3/4 AEs in 22.6% (mostly neutropenia). 1 case of grade 4 pneumonitis. No treatment-related deaths.	[134]
Nivolumab + Brentuximab Vedotin	R/R PMBCL; post-ASCT or ≥2 prior lines	PD-1 + CD30	Median PFS not reached (follow-up: 11.1 mo)	37% (investigator), 43% (indep. review)	83% had treatment-related AEs. Grade 3–4 in 53%: neutropenia (30%), thrombocytopenia (10%), peripheral neuropathy (10%). No treatment-related deaths.	[135]
Magrolimab + Rituximab	R/R indolent NHL (FL or MZL); median 3 prior lines; 65% rituximab-refractory	CD47 + CD20	Median PFS: 7.4 months; 2-year PFS rate: 27.4%	30.4%	100% had any-grade TEAEs; 95.7% treatment-related. Common: infusion reactions (60.9%), headache (52.2%), fatigue (45.7%). Grade ≥ 3: anemia (21.7%), thrombocytopenia (17.4%), neutropenia (10.9%). No treatment-related deaths.	[136]
TTI-622 (0.8–18 mg/kg weekly)	R/R lymphomas: DLBCL, CTCL-MF, PTCL, HL, FL; 27 evaluable patients (median 3 prior therapies; incl. CAR-T/HSCT)	CD47	Not reported (early-phase)	7.4% (2/27)	47% had treatment-related AEs. Most common: thrombocytopenia (21%), neutropenia (12%), anemia & fatigue (9% each). Grade ≥ 3 AEs in 5–9% range. No clear dose-related toxicity trend.	[137]

PD-L1 expression on large B-cell lymphoma cells is associated with poor outcomes due to suppression of the anti-lymphoma immune response. Consequently, targeting the PD-1/PD-L1 axis has been explored as a therapeutic strategy in R/R B-cell NHL [138]. A phase Ib trial [139] assessed nivolumab as a monotherapy in 31 patients with R/R B-cell NHL, including diffuse large B-cell lymphoma (DLBCL), follicular lymphoma (FL), and other B-cell subtypes. The patients were heavily pretreated. Nivolumab showed an ORR of 40% in FL and 36% in DLBCL, with durable responses that, in some cases, exceeded one year. There was a limited efficacy observed in NK/T-cell NHL. No objective responses were seen in other B-cell NHL subtypes. The treatment was generally well tolerated but disappointing, with most adverse events being grade 1 or 2. However, nivolumab showed promising clinical and radiographic responses in a small case series of patients with relapsed/refractory primary CNS lymphoma (PCNSL) and primary testicular lymphoma (PTL), both of which frequently harbor 9p24.1/PD-L1/PD-L2 alterations. All five patients responded to treatment, which supported further investigation of PD-1 blockade in these lymphoma subtypes [140].

The phase 2 KEYNOTE-170 study [134] evaluated pembrolizumab monotherapy in 53 patients with R/R primary mediastinal large B-cell lymphoma (PMBCL) after ≥2 prior therapies. With a median follow-up of 48.7 months, the ORR was 41.5%, including 20.8% complete and 20.8% partial responses. The median PFS was 4.3 months, and the 48-month PFS and OS rates were 33.0% and 45.3%, respectively. The responses were durable, with 80.6% lasting ≥48 months. The treatment was well tolerated. These results confirm the sustained efficacy and favorable safety profile of pembrolizumab in heavily pretreated patients with R/R PMBCL.

A phase 2 study [141] used avelumab, a PD-L1 blocking antibody, in 21 patients with R/R extranodal NK/T-cell lymphoma (ENKTL). The study found a 24% CR rate and a 38% ORR. The responses were significantly associated with high PD-L1 expression, particularly in patients with immune evasion-A (IE-A) tumor subtypes. However, the trial ended early before full enrollment due to lower-than-expected efficacy. Importantly, one patient experienced rapid disease progression, which was possibly linked to immune flare. Additionally. The ORIENT-4 phase 2 trial [142] evaluated sintilimab, a PD-1 inhibitor, in 28 patients with R/R ENKTL after failure of asparaginase-based therapy. Sintilimab achieved a 75% objective response rate (21.4% complete), with a 2-year OS rate of 78.6%.

A phase Ib/II trial [143] evaluated the triplet regimen obinutuzumab–atezolizumab–lenalidomide (G-atezo-len) in 38 patients with R/R FL. The CRR was 71.9%, with a 36-month PFS rate of 68.4% and an OS of 90%. The responses were durable and associated with a high rate of minimal residual disease (MRD) negativity. The regimen was generally well tolerated.

Combining PD-1/PD-L1 immune checkpoint blockade with CD19 CAR T-cell therapy has yielded mixed outcomes in early-phase clinical trials [144,145,146,147]. However, The Phase I Alexander study [148] evaluated a dual CD19/22 CAR T-cell therapy, combined with pembrolizumab, in patients with R/R DLBCL. The treatment showed a 64% OS rate and 55% CR rate, with no severe cytokine release syndrome or neurotoxicity.

Except for in specific NHL subtypes and, potentially, when used in combination with CAR T-cell therapy, PD-1/PD-L1 ICIs typically provides limited additional clinical benefit when combined with other active treatments in R/R NHL. Such combinations include nivolumab with ipilimumab or lirilumab, PD-1/PD-L1 inhibitors with BTK inhibitors (ibrutinib or acalabrutinib), CDK inhibitors (dinaciclib), or other agents like lenalidomide or bendamustine, with or without rituximab [130,149,150,151,152,153,154].

### 3.3. Myeloid Malignancies

The AML microenvironment contributes to immune evasion through increased levels of Tregs and immunosuppressive cytokines such as IL-10 and TGF-β, which impair effector T-cell function [155]. AML-infiltrating CD8+ T cells exhibit the silencing of genes involved in cytotoxicity and immune activation due to histone deacetylation, which reduces their responsiveness to immune checkpoint inhibition [156,157].

Intrinsic leukemic stem cells (LSCs) overexpress several immune checkpoint molecules, including PD-L1, TIM-3, and CD276, which not only suppress T-cell responses but also exert tumor-intrinsic effects that promote survival, proliferation, and stemness [155]. For instance, PD-L1 expression on LSCs enhances glycolysis via the AKT/mTOR/HIF-1α axis and reduces apoptosis, while its reverse signaling contributes to therapy resistance even without PD-1 engagement by promoting leukemic cell survival through increased fatty acid oxidation and reduced fatty acid synthesis [158,159,160]. TIM-3, highly expressed on AML LSCs but absent in healthy HSCs, drives self-renewal through an autocrine loop that involves galectin-9 and the downstream activation of the ERK, AKT, and NFκB signaling pathways [161,162].

Allogeneic hematopoietic stem-cell transplantation (allo-HSCT) remains the only curative treatment for most cases of AML and myelodysplastic syndrome (MDS), which makes it a cornerstone of therapy [163]. Its success has sparked growing interest in exploring other immunotherapeutic approaches, particularly ICIs.

Early phase 1 studies have demonstrated a limited clinical activity of anti-PD-1 monoclonal antibodies when used alone in patients with AML [164] or high-risk MDS after failure of hypomethylating agent treatment [165]. Likewise, attempts to block CTLA-4 with agents such as ipilimumab have not yielded meaningful responses in high-risk MDS [166]. However, a phase 1/1b trial [167] evaluated ipilimumab in 28 patients with hematologic malignancies that relapsed after allo-HSCT. Among those with myeloid malignancies, including extramedullary AML and MDS evolving into AML, clinical responses were observed only at the higher dose (10 mg/kg), in contrast to the prior study in non-transplanted patients, where ipilimumab at 3 mg/kg was administered without dose-limiting toxicities but failed to elicit meaningful responses. Notably, 5 of the 22 patients (23%) achieved a CR, 4 of them had extramedullary AML, and 1 had MDS/AML. Some responses were durable beyond one year. The treatment was associated with immune-related adverse events and graft-versus-host disease (GVHD), with one treatment-related death occurring. Immune analyses showed that responders had increased infiltration of cytotoxic CD8+ T cells, decreased Tregs, and elevated chemokines, such as CXCL2 and CXCL5. The findings suggest that CTLA-4 blockade may reinvigorate antitumor immunity in selected relapsed myeloid malignancy patients, particularly with extramedullary disease, although toxicity remains a concern.

Building on the above findings, a recent multicenter phase 1 trial [168] investigated the combination of ipilimumab and decitabine in patients with R/R AML or MDS, including patients who were both post-transplant and transplant-naïve. The combination demonstrated manageable toxicity and encouraging efficacy, particularly in patients who were transplant-naïve, who showed a 52% OS rate (vs. 20% post-transplant). Most responses occurred at the 10 mg/kg dose of ipilimumab, which was established as the recommended phase 2 dose. The immune-related adverse events were frequent but generally manageable, with immune activation, including CD8+ T-cell infiltration and increased ICOS expression, being seen across both arms. Notably, longer-term responders tended to exhibit immune-related toxicity, which supports the link between immune activation and clinical benefit. The combination may thus serve as a bridge to transplant or a less-intensive immunotherapeutic strategy in selected patients (Table 3).

In parallel, a multi-center phase 2 trial [174] evaluated azacitidine combined with pembrolizumab in both patients with R/R and newly diagnosed patients with AML aged ≥ 65 years who were ineligible for intensive therapy. In the R/R cohort (n = 37), the OR rate was modest (14% CR), with additional patients achieving a partial response or hematologic improvement. In contrast, the newly diagnosed cohort (n = 22) showed more promising outcomes, with a 47% CR rate and improved survival metrics. IRAEs occurred in both cohorts but they were mostly manageable with steroids in the group of patients with newly diagnosed AML, which warrants further biomarker-driven investigation to identify responders.

### 3.4. Multiple Myeloma

MM, a malignancy of plasma cells, represents about 10% of all blood cancers. MM cells actively shape a suppressive TME by upregulating immune checkpoints such as PD-1, PD-L1, CTLA-4, LAG-3, TIM-3, and TIGIT, which collectively impair T- and NK-cell function, leading to T-cell exhaustion and immune evasion [175]. This immune suppression is further reinforced by bone marrow mesenchymal stem cells which inhibit cytotoxic CD8+ T-cell activity through PD-1/PD-L1 signaling [176]. The high co-expression of checkpoints like PD-1, TIM-3, and VISTA on T cells has been observed in MM patients and is indicative of T-cell dysfunction and exhaustion [177]. Resistance is compounded by the compensatory upregulation of alternative checkpoints upon the blockade of one axis, which limits the efficacy of monotherapy [178]. Genetic variants in CTLA-4 have also been linked to increased susceptibility to MM and its precursor MGUS, which suggests a heritable component to immune dysregulation [179,180]

Major therapeutic advances such as the use of IMIDs, proteasome inhibitors, anti-CD38 antibodies, and anti-BCMA strategies like CAR-T cells and bispecific T-cell engagers, have improved outcomes, although most patients eventually develop drug resistance. ICIs were explored based on preclinical evidence which showed high PD-L1 expression in myeloma cells and CTLA-4+ regulatory T cells in the microenvironment [181,182,183]. However, clinical trials of checkpoint inhibitors in MM have largely yielded disappointing results despite this rationale.

Nivolumab monotherapy led to only 1 CR (4%) after radiotherapy, with no partial responses. Stable disease was observed in 17 patients (63%), and median the PFS was 10 weeks [139]. The phase 3 trials KEYNOTE-183 [16] and KEYNOTE-185 [184] evaluated the addition of pembrolizumab to standard immunomodulatory regimens in R/R and newly diagnosed MM, respectively. In KEYNOTE-183, pembrolizumab combined with pomalidomide and dexamethasone failed to improve the PFS compared to pomalidomide and dexamethasone alone and was associated with a higher rate of serious adverse events and treatment-related deaths, which prompted early termination. Similarly, KEYNOTE-185, which assessed pembrolizumab plus lenalidomide and dexamethasone in patients who were transplant-ineligible and treatment-naive, also showed an unfavorable risk–benefit profile, with no improvement in efficacy and increased toxicity, including more frequent immune-mediated and fatal adverse events. Both trials were halted prematurely by the FDA due to safety concerns, which highlights the limited benefit and increased risk of combining PD-1 blockade with immunomodulatory agents in MM (Table 4).

The limited clinical benefit of PD-1 blockade in MM remains unclear. MM’s growth within the bone marrow creates a distinct TME that likely contributes to immune resistance [187]. One major factor is the presence of dysfunctional, senescent T cells, which are evident even in premalignant stages and become more pronounced after ASCT, which makes T-cell reinvigoration challenging [188,189,190]. Additionally, MM progression induces the expansion of immunosuppressive cells like MDSCs and Tregs, which further impair anti-tumor immunity [191].

LAG3 and its ligand GAL-3 contribute to immune suppression in MM. Blocking either enhances T-cell proliferation and anti-MM activity, especially in memory CD8+ T cells. Anti-LAG3 is more effective and less immunosuppressive than anti-PD1, which supports its use alone or with vaccines to improve immune responses in MM [192]. The phase 1/2 MyCheckpoint trial [186] evaluated anti-LAG-3 and anti-TIGIT antibodies in combination with pomalidomide in relapsed MM patients. Among 14 participants, both regimens were well tolerated and showed promising activity, with the responses including very good partial responses and partial responses in each arm. No autoimmune toxicities were observed, and two patients demonstrated durable responses beyond 16 and 23 treatment cycles. These findings provide the first clinical evidence that LAG-3 and TIGIT blockade may help restore T-cell function and offer a novel immunotherapy approach in relapsed MM; however, a longer follow-up is warranted to confirm the durability and impact of these responses.

## 4. Mechanisms of Immune Resistance to Checkpoint Inhibitors

Figure 3 summarizes factors of immune resistance to checkpoint inhibitors.

### 4.1. Tumor-Intrinsic Mechanisms

Despite the transformative impact of ICIs in malignancies, a significant proportion of patients exhibit either no response (primary resistance) or relapse after an initial response (acquired resistance). Resistance to ICIs arises from a complex interplay of tumor-intrinsic and tumor-extrinsic mechanisms that disrupt the Cancer-immunity cycle at various stages, ultimately impairing the activation, infiltration, and cytotoxicity of T cells

A primary tumor-intrinsic factor is the lack of immunogenic neoantigens, which is often due to a low tumor mutational burden (TMB), which impairs antigen recognition by T cells. In contrast, lymphoid malignancies generally exhibit a higher TMB than their myeloid counterparts, which makes them more likely to respond to ICIs [193,194]. Acquired resistance can also arise via immune editing, whereby tumor subclones with reduced neoantigen expression are selected under immunologic pressure [195]. Additionally, mutations in antigen presentation machinery components like β2-microglobulin (β2M), MHC molecules, or TAP proteins hinder antigen processing and presentation, leading to immune evasion and resistance [196,197,198].

IFN-γ signaling exhibits a paradoxical role. While early IFN-γ signaling promotes anti-tumor immunity via the JAK-STAT-mediated upregulation of MHC I and PD-L1, prolonged signaling may lead to immune suppression and resistance by inducing the expression of inhibitory ligands or epigenetic modifications [199,200]. Loss-of-function mutations in JAK1/2 and beta-2-microglobulin (B2M) impair IFN responsiveness and contribute to both primary and acquired resistance despite eliciting a high TMB [201,202].

Epigenetic modifications such as DNA methylation and histone deacetylation alter the expression of immune-related genes, including those that encode MHC molecules and immune checkpoints. The silencing of MHC class I transactivators or CXCL9/10 impairs antigen presentation and T-cell recruitment, respectively [203,204,205]. Moreover, the nuclear translocation of PD-L1, driven by histone deacetylase activity, was shown to regulate immune response genes and mediate resistance [206].

### 4.2. Tumor-Extrinsic Mechanisms

Immunosuppressive cell populations such as Tregs, MDSCs, and M2-polarized tumor TAMs accumulate in the TME and interfere with cytotoxic T-cell activity. These cells release inhibitory cytokines (e.g., IL-10, TGF-β), express checkpoints like CTLA-4 and PD-1, and inhibit antigen presentation, collectively driving immune escape [207]. For instance, TAMs can physically block CD8+ T-cell migration and even strip PD-1 antibodies from T-cells, neutralizing therapeutic effects [208,209].

The compensatory upregulation of alternative immune checkpoints, such as TIM-3, LAG-3, and TIGIT, contributes to adaptive resistance following treatment with anti-PD-1 or anti-CTLA-4 antibodies. In various studies, the dual or triple blockade of these molecules has improved outcomes [178,202,210,211,212].

## 5. Strategies to Overcome Resistance

Figure 4 summarizes strategies to overcome resistance to checkpoint inhibitors. Overcoming resistance to ICIs is a critical focus in cancer immunotherapy, given that a significant proportion of patients fail to respond or develop acquired resistance [213].

One key approach involves combining ICIs with chemotherapy or radiotherapy, which can induce immunogenic cell death, enhance antigen presentation, and promote T-cell priming [214]. Similarly, dual immune checkpoint blockade, such as targeting PD-1 in combination with CTLA-4, LAG-3, TIM-3, or TIGIT, has shown potential in restoring anti-tumor immunity, particularly in refractory settings [210,215,216,217,218].

Reprogramming the TME is another strategy, and is aimed at reducing immunosuppressive cell populations such as Tregs and myeloid-derived suppressor cells. Agents like CCR5 and CSF1R inhibitors, along with oncolytic viruses and TLR agonists, can “heat up” the TME, improving the efficacy of ICIs [213]. Epigenetic therapies, including HDAC and DNMT inhibitors, also show promise in increasing tumor immunogenicity and MHC expression [219,220,221,222]. Personalized medicine guided by biomarkers, such as TMB, microsatellite instability, and PD-L1 expression, further refines ICI use, allowing for better patient stratification [213].

## 6. Conclusions

Immune checkpoint molecules have fundamentally reshaped the therapeutic paradigm in hematological malignancies by offering unprecedented potential to reverse immune suppression and enhance antitumor responses. While cHL demonstrates remarkable responsiveness to checkpoint inhibition, owing primarily to its characteristic amplification of PD-L1 and PD-L2, other malignancies such as NHL, AML, and MM exhibit more heterogeneous and often limited responses. This variability highlights the profound impact of TME complexity, intrinsic genomic alterations, differential checkpoint expression, and distinct immunosuppressive mechanisms.

Resistance to ICIs, whether primary or acquired, remains a significant obstacle across hematological cancers. Key tumor-intrinsic resistance factors include low TMB, impaired antigen presentation pathways, and dysregulated interferon signaling. Tumor-extrinsic factors such as immunosuppressive cell populations (e.g., regulatory T cells, myeloid-derived suppressor cells, and TAMs) and the compensatory upregulation of alternative checkpoint pathways further complicate the attainment of therapeutic success.

Addressing these challenges will require biomarker-driven and precision-based approaches that leverage comprehensive genomic and immune profiling to identify responsive patient subsets and rational therapeutic combinations. Innovative strategies—such as dual or multi-checkpoint blockade, modulation of the TME through epigenetic therapies, targeted chemotherapy-immunotherapy regimens, and optimized transplant consolidation—show promise in overcoming existing limitations.

In this context, emerging immune checkpoints such as VISTA, B7-H3, and CD47 merit increased attention. Additionally, members of the Siglec family, particularly Siglec-7, Siglec-9, and Siglec-10, have been identified as critical immunoregulatory receptors that contribute to immune escape through the suppression of natural killer (NK) cells and macrophage activity. Their therapeutic blockade may restore innate immune function and improve the efficacy of ICIs, especially in hematological malignancies where NK-cell dysfunction is prominent.

Ultimately, advancing immune checkpoint therapy in hematological malignancies demands ongoing research into novel checkpoint targets, deeper mechanistic insights into immune resistance, and the strategic incorporation of personalized immunotherapy regimens. By adopting such tailored strategies, the field can move closer to fully realizing the transformative potential of immune checkpoint inhibition across all hematologic cancers.

## Figures and Tables

**Figure 1 cancers-17-02292-f001:**
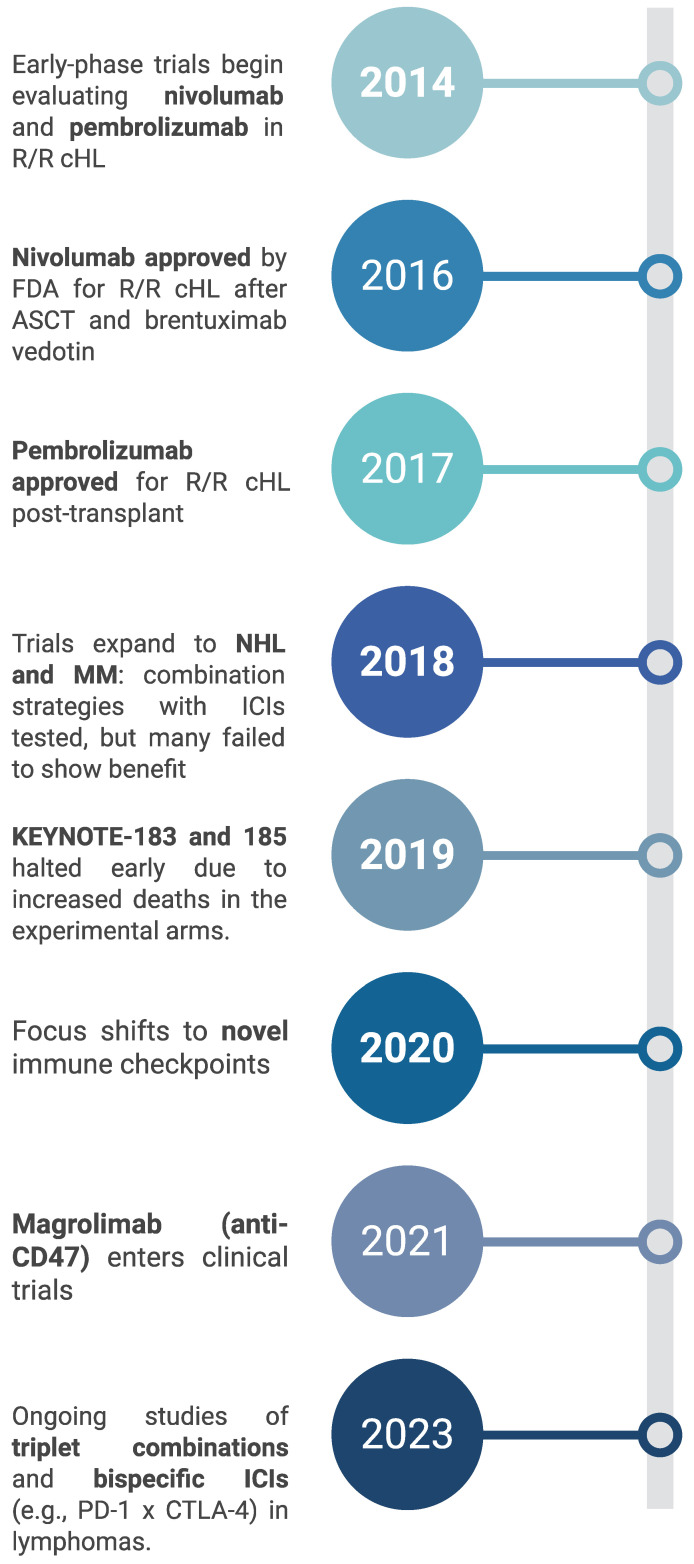
Timeline summarizing the clinical development of immunotherapy in hematological malignancies.

**Figure 2 cancers-17-02292-f002:**
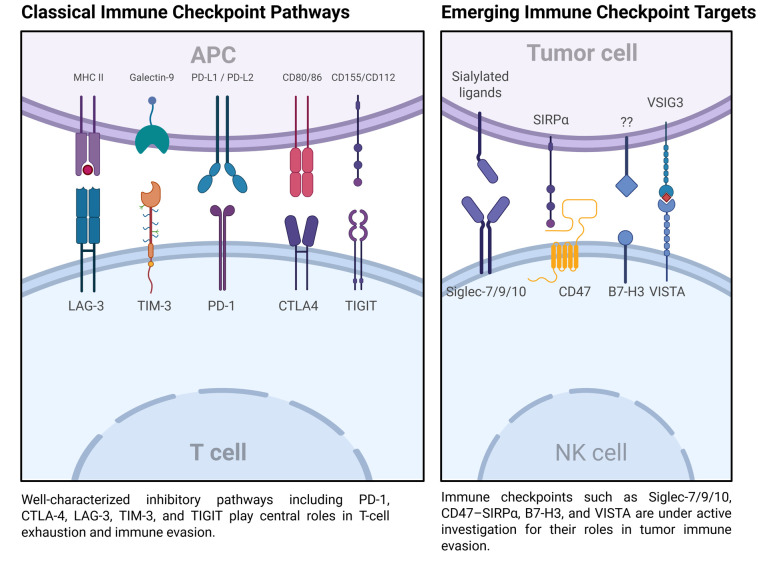
Different pathways for the mechanisms of immune checkpoint inhibitors.

**Figure 3 cancers-17-02292-f003:**
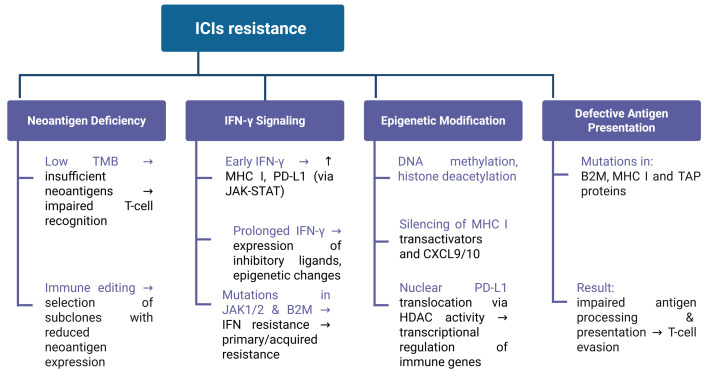
Factors of immune resistance to checkpoint inhibitors.

**Figure 4 cancers-17-02292-f004:**
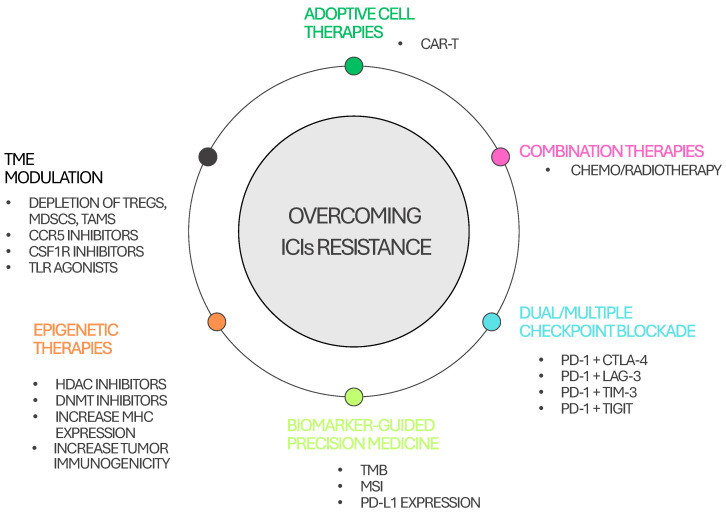
Strategies to overcome immune resistance to checkpoint inhibitors.

**Table 1 cancers-17-02292-t001:** PD1-inhibitor combination regimens for R/R Hodgkin lymphoma.

Regimen	PFS, %	CR %	Reference
BV-nivolumab	77 (3-years)	67	[118]
Nivolumab–BV ± BV–Bendamustine	91 (1-year)	59 (after induction) 94 (pre-consolidation)	[119]
nivolumab\nivolumab ± ICE	72 (2-years)	91	[120]
Pembrolizumab-ICE	87 (2-years)	87	[121]
Pembrolizumab-GVD	96 (30-months)	95	[122]

**Table 3 cancers-17-02292-t003:** Immune checkpoint inhibitors in myeloid malignancies.

Regimen	Population	Checkpoint Target	PFS	CR	Side Effects	Reference
CT-011 (0.2–6 mg/kg)	Advanced myeloid malignancies (subset)	PD-1	-	1 CR in total	Safe and well tolerated; no maximum tolerated dose reached; no major immune toxicities noted	[164]
Pembrolizumab (10 mg/kg q2w)	R/R MDS post-HMA failure (IPSS int-1 to high risk)	PD-1	Not directly reported; 24-week OS: 49%	0%	36% had treatment-related AEs. Most frequent: hypothyroidism (14%), fatigue (11%). Grade 3/4 AEs in 7% (e.g., tumor lysis syndrome, gastroenteritis). No treatment-related deaths reported.	[165]
High-dose cytarabine → Pembrolizumab 200 mg IV on Day 14 (±maintenance)	R/R AML (n = 37)	PD-1	Median OS: 11.1 months 13.2 months in R/R	CRc: 38% (composite CR) ORR: 46%	Grade ≥ 3 irAEs: 14% (rare and self-limited); generally tolerable and feasible in post-chemotherapy setting	[169]
Azacitidine (75 mg/m^2^ × 7 days/cycle) + Pembrolizumab (200 mg IV q21d)	Previously untreated, higher-risk MDS (n = 17)	PD-1	OS not reached (median FU: 13.8 mo)	3/17 (18%) CR Overall Response Rate: 80%	Most common AEs: arthralgia (40%), pneumonia (33%), nausea (27%), rash (27%); 1 early death unrelated to treatment	[170]
Azacitidine (75 mg/m^2^ Days 1–7) + Nivolumab (3 mg/kg Days 1 & 14)	Patients with relapsed AML (n = 53), median age 68; 43% secondary AML, 43% poor-risk cytogenetics	PD-1	Median OS: 5.7 mo overall; 9.3 mo in salvage-1 subgroup	CR/CRi: 21% Overall Response Rate: 35%	Grade 3/4 immune AEs in 14%, Grade 2 AEs in 12%; most resolved with steroids; 12/13 rechallenged successfully	[171]
Sabatolimab + HMA	vHR/HR-MDS (n = 53) and ND-AML (n = 48) patients	TIM-3	MDS: 12-mo PFS: 51.9% AML: 12-mo PFS: 27.9%	-	Gr ≥ 3 AEs similar to HMA alone: thrombocytopenia (~44%), neutropenia (~48%), anemia (~30%), febrile neutropenia (~33%); few immune AEs; no Gr ≥ 3 imAEs in MDS patients	[172]
Magrolimab + Azacitidine	Untreated high-risk MDS	CD47	Median PFS: 11.6 months; Median OS: not reached (17.1 mo follow-up)	33%	Most common: constipation (68%), thrombocytopenia (55%), anemia (52%). Manageable anemia (median Hb drop: −0.7 g/dL). No major immune-related AEs reported.	[173]
Ipilimumab + Decitabine	R/R or secondary MDS/AML (n = 48: 25 post-HSCT, 23 transplant-naïve)	CTLA-4	No significant OS/PFS difference between arms; 1-year OS in transplant-naïve with irAE: 72.7%	52% in transplant-naïve (5 CR, 2 CRi, 5 mCR ± HI); 20% in post-HSCT (4 CR, 1 mCR)	Grade ≥ 3 neutropenia (32–48%), thrombocytopenia (28–48%), febrile neutropenia (36–61%). irAEs in ~45%; GVHD in post-HSCT; dermatitis/colitis in transplant-naïve.	[168]

**Table 4 cancers-17-02292-t004:** Immune checkpoint inhibitors in multiple myeloma.

Regimen	Population	Checkpoint Target	PFS	CR	Side Effects	Reference
Nivolumab 1 or 3 mg/kg every 2 weeks	R/R MM (n = 27), median 3 prior therapies	PD-1	-	0%	63% had treatment-related AEs (mostly grade 1–2); stable disease in 63%. No objective responses. Only 1/14 had PD-L1+ tumors.	[139]
Nivolumab (3 mg/kg) + Ipilimumab (1 mg/kg) Q3W × 4, then Nivolumab Q2W	R/R MM (n = 7), median 5 prior therapies	PD-1 + CTLA-4	-	0%	Common AEs: fatigue (26%), pyrexia (23%), diarrhea (18%). Grade ≥ 3 AEs in 29% of patients overall. 4 MM patients died (all from disease progression). No treatment-related deaths.	[149]
Pembrolizumab monotherapy (10 mg/kg Q2W or 200 mg Q3W)	30 patients with R/R MM (median 4 prior therapies; 53% post-SCT; 67% lenalidomide-refractory)	PD-1	Median PFS: 2.7 months	0% (No CRs or PRs)	40% had TRAEs; most common: asthenia (17%), arthralgia (7%). One grade 3 TRAE (myalgia). No grade 4/5 TRAEs. 1 patient discontinued due to AE.	[185]
Pembrolizumab + Pomalidomide + Dexamethasone	125 patients with R/R MM (≥2 prior therapies, refractory to last line)	PD-1	Median PFS: 5.6 months vs. 8.4 months in control arm	-	Serious AEs in 63% (vs. 46% control); 4 treatment-related deaths: myocarditis, Stevens-Johnson syndrome, neutropenic sepsis, unknown cause. Study halted due to safety	[16]
Anti-LAG-3 (BMS-980616) or Anti-TIGIT (BMS-986207) + pomalidomide	R/R MM after ≥3 lines including anti-CD38 mAb	LAG-3/TIGIT	-	0%	Common AEs: anemia, dyspnea. Grade 3–4 AEs: dyspnea, neutropenia, thrombocytopenia. No autoimmune AEs. 1 pneumonia-related death (unrelated).	[186]

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
