# Peer review of "Immune Checkpoint Molecules in Hodgkin Lymphoma and Other Hematological Malignancies"

_cancers, 2025, doi:10.3390/cancers17142292_

Round 1
Reviewer 1 Report
Comments and Suggestions for Authors
The manuscript presents a comprehensive and timely review of immune checkpoint molecules in hematological malignancies. The authors effectively summarize key immune regulatory mechanisms and provide a detailed overview of the clinical applications and limitations of immune checkpoint inhibitors (ICIs), particularly in Hodgkin lymphoma, non-Hodgkin lymphoma, acute myeloid leukemia, and multiple myeloma. The discussion of immune resistance mechanisms and future therapeutic strategies is insightful and relevant to ongoing clinical and translational research.
To further improve the clarity and practicality of the review, the following amendments are proposed:
1) A small amount of language editing is recommended throughout the manuscript to improve sentence fluency and grammar, especially in longer paragraphs with complex ideas.
2) Consider improving the visual quality of the graphics, including clarity, color scheme, and layout, to better support and convey the key messages of the text.
3) In the section discussing mechanisms of resistance, it would be helpful to provide more specific examples that are unique to AML and MM, including known genetic alterations or microenvironmental features that contribute to immune escape.
4) Consider briefly discussing emerging checkpoint targets such as VISTA, B7-H3, or CD47 in conclusions or future directions to expand the scope of translation.
Comments on the Quality of English LanguageA small amount of language editing is recommended throughout the manuscript to improve sentence fluency and grammar, especially in longer paragraphs with complex ideas.
Author Response
|
Reviewer comments: |
Replies: |
|
1. A small amount of language editing is recommended throughout the manuscript to improve sentence fluency and grammar, especially in longer paragraphs with complex ideas. |
Thank you for the helpful suggestion. We have carefully revised the manuscript, paying particular attention to improving sentence fluency and grammar in longer paragraphs and those sections presenting complex ideas. |
|
2. Consider improving the visual quality of the graphics, including clarity, color scheme, and layout, to better support and convey the key messages of the text. |
Thank you very much for this valuable suggestion. We have fully redesigned Figure 1 (new Figure 3) to improve clarity. |
|
3. In the section discussing mechanisms of resistance, it would be helpful to provide more specific examples that are unique to AML and MM, including known genetic alterations or microenvironmental features that contribute to immune escape. |
Thank you for the suggestion. We’ve included AML and MM specific examples under the immune checkpoint mechanisms section to improve flow and clarity. |
|
4. Consider briefly discussing emerging checkpoint targets such as VISTA, B7-H3, or CD47 in conclusions or future directions to expand the scope of translation. |
Thank you for the suggestion. We have briefly included emerging targets such as VISTA, B7-H3, and CD47 in the conclusion. |
Reviewer 2 Report
Comments and Suggestions for Authors
The manuscript is well-written and informative, offering a valuable overview of immune checkpoint molecules in Hodgkin lymphoma and other hematological malignancies. The topic is timely and relevant, and the review will be helpful to both clinicians and researchers in the field. However, I have a few minor comments and suggestions that, if addressed, could further improve the clarity and completeness of the paper.
1. It would enhance the clarity and accessibility of the manuscript if a graphical abstract or schematic illustration were included in the introduction to visually represent the mechanisms of action of ICIs. Such a figure would help orient readers who may not be deeply familiar with the immunological pathways involved.
2. A brief explanation of theTME specific to each type of hematologic malignancy—such as Hodgkin lymphoma, non-Hodgkin lymphoma, multiple myeloma, and leukemias—would be highly beneficial. This would provide context for how the TME contributes to immune evasion and supports the immune-suppressive landscape, thereby justifying the therapeutic rationale for ICI use.
3. To offer a more comprehensive overview, I recommend including a dedicated discussion on immune-related adverse events associated with checkpoint inhibitors in hematologic cancer patients. This should address the types, frequency, and management of irAEs in this population, and consider whether their presentation differs from that observed in solid tumors.
4. In addition to PD-1/PD-L1, LAG3, TIM and CTLA-4, it would strengthen the review to include a discussion on emerging immune checkpoint pathways, such as CD47/SIRPα and VISTA, which are currently under active investigation in hematologic malignancies. Moreover, Siglec family members, particularly Siglec-7, Siglec-9, and Siglec-10, are increasingly recognized as important immunomodulatory receptors that contribute to immune evasion in hematologic tumors. Their involvement in NK cell and macrophage inhibition, and the potential for therapeutic targeting, warrants inclusion in this review.
Author Response
|
Reviewer comments: |
Replies: |
|
1. It would enhance the clarity and accessibility of the manuscript if a graphical abstract or schematic illustration were included in the introduction to visually represent the mechanisms of action of ICIs. Such a figure would help orient readers who may not be deeply familiar with the immunological pathways involved. |
Thank you for the feedback. We have included a schematic illustration to visually represent the mechanisms of action of ICs (new Figure 2). |
|
2. A brief explanation of the TME specific to each type of hematologic malignancy—such as Hodgkin lymphoma, non-Hodgkin lymphoma, multiple myeloma, and leukemias—would be highly beneficial. This would provide context for how the TME contributes to immune evasion and supports the immune-suppressive landscape, thereby justifying the therapeutic rationale for ICI use. |
Thank you for the suggestion. We’ve included AML and MM specific examples under the immune checkpoint mechanisms section to improve flow and clarity. Regarding the specific Reviewer’s suggestion a brief explanation of the TME specific to each type of hematologic malignancies can be found in the revised manuscript (section 3.1 Hodgkin lymphoma; 3.2 Non Hodgkin lymphoma; 3.3 Leukemias; and 3.4 multiple myeloma. |
|
3. To offer a more comprehensive overview, I recommend including a dedicated discussion on immune-related adverse events associated with checkpoint inhibitors in hematologic cancer patients. This should address the types, frequency, and management of irAEs in this population, and consider whether their presentation differs from that observed in solid tumors. |
Thank you for the suggestion. We have now included a dedicated section discussing immune-related adverse events |
|
4. In addition to PD-1/PD-L1, LAG3, TIM and CTLA-4, it would strengthen the review to include a discussion on emerging immune checkpoint pathways, such as CD47/SIRPα and VISTA, which are currently under active investigation in hematologic malignancies. Moreover, Siglec family members, particularly Siglec-7, Siglec-9, and Siglec-10, are increasingly recognized as important immunomodulatory receptors that contribute to immune evasion in hematologic tumors. Their involvement in NK cell and macrophage inhibition, and the potential for therapeutic targeting, warrants inclusion in this review. |
Thank you for the insightful comment. We have addressed this in the future perspectives section. The revised manuscript includes, in addition to a discussion on emerging immune checkpoint pathways which are currently under active investigation, important immunomodulatory receptors that contribute to immune evasion in hematologic tumors. |
Reviewer 3 Report
Comments and Suggestions for Authors
In this manuscript, the authors reviewed the mechanisms of immune checkpoint regulation and the clinical applications of immune checkpoint inhibitors in hematological malignancies. It also discussed emerging strategies to overcome immune checkpoint inhibitor resistance, including dual checkpoint blockade and tumor microenvironment modulation. The topic fits the scope of this journal and may benefit the precision-based, tailored immunotherapy across hematological malignancies. In general, this manuscript is well-organized and the references are updated. Key issues are required to be addressed before its publication in Cancers.
Major points:
No.
Minor points:
- It is recommended to introduce the immune checkpoint pathways using one or more illustrative figures to enhance clarity and improve overall readability for the audience.
- Including a timeline or milestone figure summarizing the clinical development of immunotherapy would provide helpful context and strengthen the manuscript’s structure.
- A brief discussion on the future directions or emerging trends in the development of immune checkpoint inhibitors is recommended to offer a forward-looking perspective.
Author Response
|
Reviewer comments: |
Replies: |
|
Thank you for the feedback. We have included a schematic illustration to visually represent the mechanisms of action of ICs (new Figure 2). |
|
2. Including a timeline or milestone figure summarizing the clinical development of immunotherapy would provide helpful context and strengthen the manuscript’s structure. |
Thank you for the suggestion. We have added a new figure (Figure 1) summarizing key milestones in the clinical development of immunotherapy. |
|
3. A brief discussion on the future directions or emerging trends in the development of immune checkpoint inhibitors is recommended to offer a forward-looking perspective.
|
Thank you for the insightful comment. We have addressed this in the future perspectives section |